# Step-by-Step Double-Trouble OBAIRH and DMD Diagnosis in a One-Year-Old Boy

**DOI:** 10.3390/ijms241512357

**Published:** 2023-08-02

**Authors:** Olga Shchagina, Vera Kurilova, Elena Zinina, Vyacheslav Porubov, Svetlana Efishova, Aleksander Polyakov

**Affiliations:** 1Research Centre for Medical Genetics, Moscow 115522, Russia; zininalen@yandex.ru (E.Z.); apol@dnalab.ru (A.P.); 2State Budgetary Institution of Health of the Perm Region “Regional Children’s Clinical Hospital”, Perm 614066, Russia; podkb_mgk@mail.ru (V.K.); porubov.vs@gmail.com (V.P.); efishova-s@mail.ru (S.E.)

**Keywords:** OBAIRH, DMD, double trouble, *POMC*

## Abstract

We present a case of a combination of two rare hereditary disorders: obesity, adrenal insufficiency and red hair syndrome (OBAIRH) and Duchenne muscular dystrophy (DMD) in a boy. Both diseases were diagnosed during the first year of life. OBAIRH was suggested based on the ethnicity and family history of the patient, while DMD was based on an extreme increase in transaminase and CK (creatine kinase) levels during a biochemical analysis of his blood. The OBAIRH syndrome was caused by a pathogenic homozygous variant in the regulatory region of the *POMC* gene (NM_001035256.3): c.-71+1G>A, while DMD was caused by the de novo deletion of exons 38–45 of the *DMD* (NM_004006.3) gene (NC_000023.10:g.(?_32380941)(31950285_?)del).

## 1. Introduction

A large number of hereditary disorders are congenital, noticeable immediately after birth, with some of them even noticeable during ultrasound examination at various terms of pregnancy. However, most diseases have no distinct phenotypical features. Diagnosis is often established after the manifestation of pronounced clinical symptoms, when therapy is ineffective. Some hereditary disorders are included in the mass neonatal screening programmes, which allow for the formation of a risk group, confirmation of diseases, and for therapy to start before the manifestation of symptoms. The inclusion of a disorder into such programmes is considered in the Wilson and Jungner Criteria; candidates for disease screening are considered if they meet the following principles: (1) the condition should be an important health problem; (2) there should be an accepted treatment for patients with the disease; (3) facilities for diagnosis and treatment should be available; (4) there should be a recognizable latent or early symptomatic stage; (5) there should be a suitable test or examination; (6) the test should be acceptable to the population; (7) the natural history of the condition, including development from latent to declared disease, should be adequately understood; (8) there should be an agreed policy regarding whom to treat as patients; (9) the cost of casepfinding (including diagnosis) should be economically balanced in relation to possible expenditure on medical care as a whole; (10) case-finding should be a continuing process and not a “once and for all” project. Therefore, rare genetic disorders are often left undiagnosed during the infant’s first year of life. Cases of diagnosing two rare hereditary disorders, which are not included in the screening, in one patient in the first year of life are extremely rare. We present a case of a combination of two rare hereditary disorders: obesity, adrenal insufficiency and red hair syndrome (OBAIRH) and Duchenne muscular dystrophy (DMD) in a boy. Both diseases were diagnosed during the first year of life. OBAIRH was suggested based on the patient’s ethnicity and family history, while DMD was based on the extreme increase in transaminase and CK (creatine kinase) levels during biochemical blood analysis.

## 2. Results

Patient D. was initially referred to a geneticist at the age of five months. He was hospitalized in the pediatric unit of the “Regional children’s clinical hospital”, Perm Russia, with an atonic seizure and cyanosis as a complication of a feverless viral infection. The anamnesis showed that, in the early neonatal period, the patient had a seizure because their hypoglycemia dropped to 1.1 mmol/L (norm is 3.3–5.5 mmol/L) without any registered epileptic activity. The seizures did not repeat until the present episode: in the hospital, during the preparation for brain MRI (magnetic resonance imaging), as a result of prolonged fasting, the patient had another atonic seizure with the glucose level decreasing to 1.1 mmol/L.

The child was born from second pregnancy, first birth, at 40 weeks; weight was 3630 g, length 51 cm, head circumference 36 cm, NS Apgar score 6/8. His parents were healthy; the father was 35 years old and the mother was 21 years old; consanguinity was denied. Both parents belonged to an ethnographic group of Perm’ Tatar—a Tatar group included in the Kazan’ Tatar group and inhabiting the territory of the Perm’ ethnocultural region [1]. The Perm’ Tatar group is the second-largest ethnic group of the Perm’ Krai [2].

The results of the examination at the age of five months were unavailable; at the age of one year, the height was 84 cm (greater than the 97th percentile), weight 16.5 kg (greater than the 97th percentile), head circumference 54 cm (greater than the 97th percentile), and thoracic circumference 69.5 cm (greater than the 97th percentile). The skin was clean (skin rash and atypical pigmentation was absent), the hair was red, no pigmentation defects were detected. Thoracic shape was normal, joints unaltered; abdomen was soft and painless; liver and spleen size were normal; stool and diuresis were normal. He had male genitals; microgenitalism; testicles were not in the scrotum. The patient did not sit up on his own but sat independently while seated, stood and walked with support, babbled, and followed toys with his eyes.

Genealogical analysis showed that the proband’s cousin on the mother’s side was affected by the obesity, adrenal insufficiency, and red hair syndrome (OMIM 609734) due to a POMC (proopiomelanocortin) deficiency caused by a homozygous ethnic c.-71+1G>A mutation in the regulatory region of the *POMC* gene (Figure 1). This option was not available in the gnomad database.

Blood hormone level analysis at the age of five months showed low cortisol levels of <13.8 nmol/L (reference levels 101.2–535.7 nmol/L), while ACTH (adrenocorticotropic hormone), insulin, TSH (thyroid - stimulating hormone), and T4 (thyroxine free) levels were normal.

Biochemical examination at the age of five months detected a significant elevation in transaminase levels AST (aspartate aminotransferase)—374 MU/L (reference < 40 MU/L), ALT(alanine aminotransferase) —424 MU/L (reference < 40 MU/L), CK—14,333 U/L (reference 24–190 U/L), CK-MB (creatine kinase-MB) —261 U/L (reference 0–24 U/L), LDH (lactate dehydrogenase) —1594 U/L (reference 225–937 U/L); total protein, albumin, creatinine, urea, alkaline phosphatase, CRP (c reactive protein), bilirubin, glucose (without a seizure), and electrolytes were normal. At the age of one year, AST—228 MU/L (reference < 40 MU/L), ALT—197 MU/L (reference < 40 MU/L), CK—10,005 U/L (reference 24–190 U/L), CK-MB—253 U/L (reference 0–24 U/L), LDH—1633 U/L (reference 225–937 U/L); total protein, albumin, creatinine, urea, alkaline phosphatase, CRP, bilirubin, glucose, and electrolytes were normal (Table 1).

Ultrasound examination of the abdominal cavity at the age of five months and one year showed no structural alterations in the organs. Brain MRI at the age of five months detected no focal structural alterations in the brain; however, the subarachnoid space was expanded. Ultrasound imaging of the heart showed a functioning oval window.

Based on the genealogical analysis results, the patient’s phenotype (excessive body mass and height, red hair), hypoglycemia episodes, a decrease in cortisol level, and the ethnicity of Perm’ Tatar, patient D. was diagnosed with OBAIRH, which was confirmed by molecular genetic analysis—Sanger sequencing of the regulatory region of the *POMC* gene (NM_001035256.3): a pathogenic c.-71+1G>A variant was detected in a homozygous state.

However, biochemical blood analysis for the proband’s cousin with the same symptoms and genotype did not show any alterations: AST, ALT, and CK levels were within the reference values.

Seeing that patient D. did not have hepatic pathology, a primary muscle lesion was suggested. At the age of ten months, electromyography of the shoulder girdle and lower limbs was carried out. This detected a significant decrease in MUAP duration and amplitude, which may signify muscular lesion type; however, the examination was not completed due to the patient’s anxiety.

Molecular genetic analysis of the *DMD* gene (NM_004006.3) using quantitative MLPA detected a hemizygous deletion spanning exons 38–45 of this gene NC_000023.10:g.(?_32380941)(31950285_?)del and leading to a frameshift.

Molecular genetic analysis was also carried out for the proband’s relatives. His mother, grandmother, and sister did not carry this deletion. A heterozygous *POMC* (NM_001035256.3) variant c.-71+1G>A was detected in the proband’s parents, aunt and her husband (parents of the patient’s cousin with OBAIRH), grandfather, and uncle. The grandmother did not have the c.-71+1G>A variant.

The medication for obesity therapy in case of proopiomelanocartin deficiency (Setmelanotide) cannot be prescribed until the age of six months. For patients with DMD caused by a deletion of exons 38–45 of the *DMD* gene, pathogenetic and etiotropic therapy is unavailable; however, corticosteroids are proven to be effective in patients with this type of muscular dystrophy [3] as a supportive therapy during the wait for new genotherapeutic medication for this disorder [4].

## 3. Discussion

The OBAIRH syndrome is an extremely rare hereditary disorder with autosomal recessive inheritance type [5]. However, patient D. belongs to an ethnic group of Perm’ Tatar, for which the accumulation of this disease, caused by the c.-71+1G>A variant’s spread in this group due to the founder effect [6], was shown; in addition, there was a case of this syndrome in the proband’s cousin. The accumulation of ethnically specific recessive mutations caused by the founder effect was described for various populations in the Russian Federation: osteopetrosis caused by the c.807+5G>A mutation in the *TCIRG1* gene, as well as erythrocytosis caused by the c.598C>T variant in the *VHL* gene [7,8,9] in Chuvash and Mari El populations, hyperphenylalaninemia caused by the p.Arg261* variant in the *PAH* gene and type 1 tyrosinemia caused by the c.1025C>T (Pro342Leu) mutation in the *HT1* gene in North Caucasian populations [10], methemoglobinemia caused by the c.806C>T variant in the *CYB5R3* gene in the Yakut group [11], as well as other disorders [12]. Thus, during medical genetic counseling, information about the patients’ ethnic origin and studies on the accumulation of rare genetic variants in certain ethnic groups are crucial. This information can shorten the patient’s path to definitive diagnosis and corresponding adequate therapy. The c.-71+1G>A variant in the *POMC* gene (NM_001035256.3) is located in the splicing region of a non-coding exon; however, despite the fact that the mechanism of its pathogenicity is not connected to the formation of aberrant RNA (ribonucleic acid) forms, it still leads to a drastic decrease in RNA and POMC protein levels, which could be connected to the decrease in transcription levels, or to the nuclear degradation of pre-mRNA [6].

The combination of two genetic disorders is quite rare, and usually one of the diseases has a high prevalence in the population. Duchenne muscular dystrophy is very common: according to various sources, the frequency is 1 in 3500–10,000 males [13,14]. Some studies describe combinations of DMD caused by deletions in the *DMD* gene with skeletal dysplasia: osteogenesis imperfecta (mutations in the *COL1A1* gene) and pseudoachondroplasia (mutation in the *COMP* gene) [15]. A Czech patient had a phenotype caused by the combination of a point mutation in the *DMD* gene and a duplication of the *PMP22* gene causing CMT1A (Charcot-Marie-Tooth disease type 1A) [16], and a Chinese patient had a phenotype caused by a duplication of the *PMP22* gene and a *DMD* deletion [17]. In both cases, CMT1A was inherited by families from generation to generation, and DMD was caused by de novo mutations. There are described cases of combinations of various muscular dystrophy forms, such as Becker muscular dystrophy, inherited from a carrier mother, and facioscapulohumeral muscular dystrophy, inherited from the father [18].

Duchenne muscular dystrophy is usually diagnosed in male patients aged from three to five years, when neurologists note progressive muscular weakness and gait impairments [3]. Detecting DMD at such a young age (one year) was possible due to the diagnostic apprehension of the doctors, who noted additional symptoms that were not characteristic of OBAIRH. A constant increase in transaminase levels, which was not detected in the proband’s relative with POMC deficiency, was a reason to search for a second disorder. As a result, we detected the de novo deletion of exons 38–45 of the *DMD* gene, leading to a frameshift, which is characteristic of a more severe disease form: Duchenne muscular dystrophy.

When the diagnosis was established, replacement therapy using glucocorticosteroids was implemented: the dose for this age was 1.25 mg, three times per day. Fractional nutrition was prescribed every 2–3 h, with the exception of nighttime. The glucose levels underwent constant monitoring and were found to be within the norm (3.5–4.5 mmol/L); they did not decrease after a night’s sleep.

The use of NGS (next generation sequencing) methods in diagnostic practice leads to the identification of many secondary findings. It is not always possible to find clinical manifestations of such findings, and the question of what to do with them in real clinical practice remains unanswered. However, in this case, a targeted diagnosis was carried out based on clinical and biochemical markers, which made it possible to identify two hereditary diseases at a very early age of the patient.

## 4. Materials and Methods

DNA was extracted from whole blood samples using a Wizard^®^ Genomic DNA Purification Kit (Promega, Madison, WI, USA) according to the manufacturer’s protocol.

Automated Sanger sequencing was carried out using an ABIPrism 3500xl Genetic Analyzer (Applied Biosystems, Foster City, CA, USA) according to the manufacturer’s protocol. Primer sequences were chosen according to the *POMC* (NM_001035256.3) reference sequence. *POMC* fragment sequences were amplified using the patient’s and relatives’ genomic DNA as a template, with the following primers: POMCF:TAGGGCAAGCGGCGGCGAAGGAGG and POMCR:TTCGCACGATCTCGGCATCTTCCAG.

Quantitative analysis was carried out using the SALSA MLPA Probemix P034-DMD1 and P035-DMD2 (MRC-Holland, Amsterdam, The Netherlands).

This study was conducted according to the guidelines of the Declaration of Helsinki and approved by the Institutional Review Board of the Research Center for Medical Genetics, Moscow, Russia (approval number 2018-3/4). The probands gave informed consent to the genetic testing and the publication of anonymized data.

## Figures and Tables

**Figure 1 ijms-24-12357-f001:**
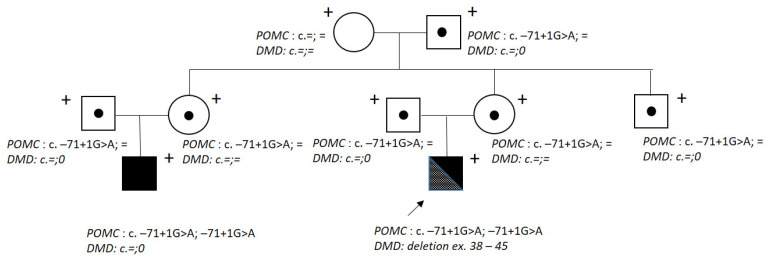
The proband’s family tree. The arrow points to patient D. Plus signs indicate the examined and genotyped family members; dots—carriers of the c. –71+1G>A variant in the *POMC* gene; black colour—patients with OBAIRH. =: a sequence was found unchanged, 0: absence of the second X chromosome (hemizygote).

**Table 1 ijms-24-12357-t001:** Tests results outside the limits of reference values.

Patient’s Age		Five Months	One Year
Index (Unit)	Reference Values		
AST (MU/L)	<40	374 MU/L	228 MU/L
ALT (MU/L)	<40	424 MU/L	197 MU/L
CK (U/L)	24–190	14,333 U/L	10,005 U/L
CK-MB (U/L)	0–24	261 U/L	253 U/L
LDH (U/L)	225–937	1594 U/L	1633 U/L
Cortisol levels (nmol/L)	101.2–535.7	<13.8 nmol/L	<13.8 nmol/L
Glucose levels (mmol/L)	3.5–4.5	3.79 nmol/L	4.1 nmol/L

## Data Availability

We can provide them or conduct a reanalysis upon request by mail schagina@med-gen.ru. Raw data cannot be posted due to patient confidentiality.

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
