# Peer review of "Step-by-Step Double-Trouble OBAIRH and DMD Diagnosis in a One-Year-Old Boy"

_ijms, 2023, doi:10.3390/ijms241512357_

Round 1

Reviewer 1 Report

The manuscript is a  case report of a one-year old boy with both OBAIRH and Duchenne muscular dystrophy. Both disorders were confirmed by molecular analysis. This is an interesting case and shows the importance of obtaining family history and ethnicity and comparing the phenotype and genotype with family members.

Please address the following:

Per OMIM (Online Mendelian Inheritance in Man) the phenotype is called obesity, adrenal insufficiency, and red hair (OBAIRH)(phenotype MIM number: 609734). The order is incorrect in the abstract and introduction sections.

Lines 27-29 of the Introduction section state NBS programs screen disorders based on frequency and availability of effective therapy. However, this needs to be expanded to the Wilson and Jungner criteria for screening.

The writing of the manuscript needs attention. For example, manuscripts should be written in the past tense but there are multiple sentences that are written in the present tense:

e.g. the anamnesis shows that in; His parents are healthy; the father is 35 years old; etc.

Line 42 – please include the normal range for glucose. Also, low glucose instead of “down to”.

Line 46  -“at the term of 40 weeks” is incorrect English.

Please clarify: “the skin was clean” (line 55)

This is not a complete sentence: “Male genitals, microgenitalism, testicles not in scrotum.” “Scrotum” is incorrectly spelled. (line 58)

(line 65) “Blood hormone level analysis at the age of five months showed a decrease in cortisol levels”. A decrease from what? Do you mean cortisol levels were low? Also line 58: do you mean transaminiase levels were elevated or they had increased?

Line 71 and 75: “albumin” rather than “albumins”.

It would be helpful if test results and ranges were tabulated.

Lines 108 to 113 regarding treatment options should be moved to the Discussion section as they are not part of the results.

English language needs to be improved.

Author Response

We express our great gratitude to Reviewer 1 for the careful study of the text of the manuscript and valuable comments. Corrections have been made to the text. After the changes were made, the article underwent professional proofreading to correct errors in the translation.

Please address the following:

Per OMIM (Online Mendelian Inheritance in Man) the phenotype is called obesity, adrenal insufficiency, and red hair (OBAIRH)(phenotype MIM number: 609734). The order is incorrect in the abstract and introduction sections.

Thank you very much for your comment. We apologize for our inattention. Fixed (highlighted in yellow in the text)

Lines 27-29 of the Introduction section state NBS programs screen disorders based on frequency and availability of effective therapy. However, this needs to be expanded to the Wilson and Jungner criteria for screening.

Thanks for the comment. Corrected in the text

The writing of the manuscript needs attention. For example, manuscripts should be written in the past tense but there are multiple sentences that are written in the present tense:

e.g. the anamnesis shows that in; His parents are healthy; the father is 35 years old; etc.

Fixed (highlighted in yellow in the text)

Line 42 – please include the normal range for glucose. Also, low glucose instead of “down to”.

Changed, thanks, but 1,1 mmol/l, this is the limit of the measuring device, so the glucose level is indicated in this format.

Line 46  -“at the term of 40 weeks” is incorrect English

Fixed (highlighted in yellow in the text)

Please clarify: “the skin was clean” (line 55)

Updated

This is not a complete sentence: “Male genitals, microgenitalism, testicles not in scrotum.” “Scrotum” is incorrectly spelled. (line 58)

Fixed (highlighted in yellow in the text)

(line 65) “Blood hormone level analysis at the age of five months showed a decrease in cortisol levels”. A decrease from what? Do you mean cortisol levels were low?

Fixed (highlighted in yellow in the text)

Also line 58: do you mean transaminiase levels were elevated or they had increased?

They were elevated. Thanks for the comment. Fixed (highlighted in yellow in the text)

Line 71 and 75: “albumin” rather than “albumins”.

Fixed (highlighted in yellow in the text)

It would be helpful if test results and ranges were tabulated.

It's a great idea! Fixed (highlighted in yellow in the text)

Lines 108 to 113 regarding treatment options should be moved to the Discussion section as they are not part of the results.

Fixed (highlighted in yellow in the text)

Reviewer 2 Report

Shchagina et al. presented a case with OBAIRH and DMD. Overall, it is well-written and concise. These are a few minor comments.

  1. Give a proper HGVS nomenclature for the DMD frameshift variant.
  2. Please check the frequency of the splice variant in gnomAD and mentioned it in the text as well.
  3. Please clearly define what "=" and "0" sign represent? and describe in the figure legend or it can be change to POMC+/+ for wildtype, POMC+/- for heterozygous,  POMC-/- for homozygous. Whereas DMDY/+ for wildtype and DMDY/- for the hemizygous variant.
  4. Is there any limitation of this study, if yes please describe it in the discussion.

Author Response

Thanks to the reviewer 2. All necessary corrections have been made to the text of the manuscript

Comments and Suggestions for Authors

Shchagina et al. presented a case with OBAIRH and DMD. Overall, it is well-written and concise. These are a few minor comments.

  1. Give a proper HGVS nomenclature for the DMD frameshift variant.

Thanks! Fixed (highlighted in green in the text)

  1. Please check the frequency of the splice variant in gnomAD and mentioned it in the text as well.

Thanks! Fixed (highlighted in green in the text)

  1. Please clearly define what "=" and "0" sign represent? and describe in the figure legend or it can be change to POMC+/+ for wildtype, POMC+/- for heterozygous,  POMC-/- for homozygous. Whereas DMDY/+ for wildtype and DMDY/- for the hemizygous variant.

Thanks!  Fixed (highlighted in green In the caption to the picture)

  1. Is there any limitation of this study, if yes please describe it in the discussion.

Thanks! Fixed (highlighted in green in the text)